# Mini Review: Co-Existing Diseases and COVID-19—A One Way Ticket?

**DOI:** 10.3390/ijerph19084738

**Published:** 2022-04-14

**Authors:** Mantė Eidininkienė, Jelena Cesarskaja, Simona Talačkaitė, Vilma Traškaitė-Juškevičienė, Andrius Macas

**Affiliations:** 1Medical Academy, Faculty of Medicine, Lithuanian University of Health Sciences, LT-44307 Kaunas, Lithuania; jelena.cesarskaja@gmail.com (J.C.); talackaites3@gmail.com (S.T.); vilma.traskaite-juskeviciene@kaunoklinikos.lt (V.T.-J.); andrius.macas@kaunoklinikos.lt (A.M.); 2Department of Anesthesiology, Hospital of Lithuanian University of Health Sciences Kaunas Clinics, LT-50160 Kaunas, Lithuania

**Keywords:** SARS-CoV-2, coronavirus, comorbidities, cardiovascular, metabolic syndrome, immunosuppression, hematologic, nephrological

## Abstract

**Background and aims**. Many patients with SARS-CoV-2 virus infection have various comorbidities. Their presence in the background of coronavirus has a tendency to worsen the course of the disease and increase the risk of unfavorable outcomes. Understanding the interactions between SARS-CoV-2 and the most common comorbidities is key to the successful management of these patients. **Methods**. We systematically searched Medline, Springer and Elsevier databases and accessed the full text on SARS-CoV-2 virus infection and the following conditions: cardiovascular, renal, immunosuppression, metabolic disorder and hematological in order to prepare a narrative review on this topic. **Results**. Patients with underlying cardiovascular diseases are more likely to suffer from severe forms of COVID-19. Cardiovascular diseases were also noted as the most frequent comorbidities among coronavirus patients. Metabolic syndrome and its components have been identified as the second most common comorbidity among fatal cases of COVID-19. Infected patients with acute kidney injury also show a higher mortality rate among the others. Immunocompromised patients, such as organ recipients and cancer and hematologic patients, develop more severe forms of COVID-19 and are at higher risk of admission to ICUs and requiring mechanical ventilation. Higher mortality rates among those patients have also been observed. **Conclusions**. Based on recent studies, patients with co-existing diseases are at higher risk for severe courses of COVID-19 virus infection and unfavorable outcomes. Cardiovascular diseases, metabolic syndrome and immunosuppressive and kidney diseases in the presence of coronavirus may lead to longer and more aggressive treatment in the ICU and increased mortality rate.

## 1. Introduction

From Wuhan Market in China (a place linked to the first cases of COVID-19), to Lithuania (7233 km further), SARS-CoV-2 virus infection has become an unprecedented challenge for healthcare providers all over the world. As at the date of writing this manuscript, COVID-19 was declared a global pandemic that had infected over 430 million people and caused more than 5.9 M deaths [1]. Despite the quite successful management during the first wave, the second wave of the pandemic in Lithuania turned into a tsunami, reaching over 2000 new COVID-19 cases per day with charts showing a spike in virus-related deaths. As of today, SARS-CoV-2 infection has caused the death of 8482 Lithuanians [2]. Our university hospital physicians who have worked directly in COVID-19-dedicated wards and intensive care units have observed that in most cases, pre-existing comorbidities play a huge role in the patient’s course of disease and unfavorable outcomes. Some prior diseases and conditions (for instance, cardiovascular, nephrological, immunosuppression, metabolic disorders, etc.) seem to be predominant in worsening the course of COVID-19 more than others. Our aim was to review the role of certain underlying comorbidities in COVID-19 patients regarding their influence on unfavorable and fatal outcomes.

## 2. Materials and Methods

The expanded literature search was developed by the PRISMA screening system protocol (Preferred Reporting Items for Systematic Reviews and Meta-Analyses). A protocol for the analysis and data collection of the publication was prepared, according to which scientific articles were selected. For scientific publications we used Medline (PubMed), Springer (SpringerLink) and Elsevier (ScienceDirect) databases using combined terms: “novel virus”, “COVID-19”, “SARS-CoV-2”, “corona virus”, “novel coronavirus”, “2019 coronavirus”, “coronavirus”, “novel COVID-19” and “COVID virus” together with “cardiovascular diseases”, “heart”, “circulatory system”, “blood vessel”, “blood stream”, “venous blood system” or “metabolic diseases”, “obesity”, “overweight”, “diabetes”, “insulin-dependent diabetes”, “pancreatic diabetes”, “diabetic patient” or “immunosuppression”, “cancer”, “malignancy”, “tumor, “carcinoma”, “immunocompromised” or “hematologic patients” or “kidney diseases”, “kidney”, “renal”, “renal disease”, “nephritic” or “age”, “adult”, “teen”, “grown-up”, “juvenile”, “adolescent”, “minor”, “youth” or “man”, “women”, “male” and “female”. Case reports, clinical analysis and fundamental research debating the influence of the novel virus to our body system were collected. Data were filtered and narrowed down to information regarding COVID-19 interference with cardiovascular, renal, blood and endocrine system comorbidities. Analyzing the cases with clinical diagnosis of disease we also combined patients of different sexes and ages, in order to clarify the differences in outcomes in those populations. The reference search and selection flow are presented in Figure 1. The summary of the studies included in this mini review is presented in Table 1. The key findings of selected studies are presented and discussed in relation to organ systems in the following paragraphs: “Immunosuppression and COVID-19“, “Cardiovascular diseases and COVID-19”, “Metabolic syndrome and COVID-19”, “Hematologic diseases and COVID-19”, “Kidney diseases and COVID-19”, “Unmodified factors and COVID-19”. The summary of the main results related to comorbidities is presented in Table 2.

## 3. Results and Discussion

### 3.1. Immunosuppression and COVID-19

Nowadays, more than two years into the pandemic, it is obvious that patients suffering from congenital or acquired immunosuppressive states are at great risk of unfavorable outcomes due to COVID-19 infection. A cohort study by Liu et al. [3] indicated a significantly higher percentage of infected patients with the final, 4th stage of cancer. They also made insights about the different and more severe illness course in patients with cancer diagnoses. Unsurprisingly, cancer patients or cancer survivors are more likely to be admitted to ICUs, receive invasive ventilation and experience lethal outcomes. Patients who already had cancer before being infected with the novel virus show a higher probability of severe events [4].

Another group in a lifelong immunocompromised condition is organ recipients. A combined group of 18 virus-infected patients with kidney, liver or heart transplants was studied in Spain by the University Hospital of Madrid. A total of 25% of the patients presented respiratory deficiency and 83.3% of them had fever at the beginning of hospitalization. During their time in hospital, chest radiography was performed and 13 out of 18 patients had abnormalities in their lungs caused by COVID-19. At the end of the study, two patients were transferred to the ICU requiring advanced respiratory management such as invasive mechanical ventilation and five patients died [6]. Moreover, comorbidities such as diabetes may be a threatening condition in the life of a kidney recipient. In a British report of seven cases, it was observed that all three patients who had the most severe disease process also had diabetes diagnosed earlier in life [7]. Transplant recipients are at higher risk of novel virus infection not only due to their immunocompromised state but also because of the comorbidities. 

### 3.2. Cardiovascular Diseases and COVID-19 

The ongoing pandemic has shown that COVID-19 virus has diverse cardiovascular clinical manifestations. Regardless of that, in the early stages of this outbreak, epidemiologists made significant observations that, compared to the general population, patients with co-existing cardiovascular diseases were more likely to suffer from severe forms of COVID-19 [7]. Recent literature findings confirm these observations. The largest summary of case reports published by the Chinese Center for Disease Control and Prevention, consisting of 44,672 patient-confirmed cases of COVID-19, indicates that the case-fatality rate was elevated in the presence of other heart diseases or hypertension at 10.5% and 6.0%, respectively, while the overall case-fatality rate was 2.3% [7]. Furthermore, an Italian retrospective case series of 1591 patients from the Lombardy region proclaimed that hypertension was the most common comorbidity among COVID-19 patients admitted to the ICU (509 patients), whereas cardiovascular diseases remained in the second place (233 patients) [8]. The study also showed that hypertensive patients were older and had higher PEEP levels and lower PaO_2_/FiO_2_, not to mention the higher numbers of fatal outcomes, compared with those discharged from the ICU. The collected data of another early study from Wuhan, China, presented the same pattern among the infected patients, where hypertension and cardiovascular diseases were the second most common underlying diseases worsening their overall condition [9].

The question of why pre-existing cardiovascular pathologies have such an enormous impact on COVID-19 patients’ outcomes remains debatable. There are several theories that partially explain this connection, starting from the general understanding that acute respiratory infections of viral and bacterial etiologies can aggravate existing cardiac disease or trigger new cardiac events [10]. Another considerable role in the interaction between a virus and the presence of cardiovascular history belongs to the angiotensin-converting enzyme 2 (ACE 2) pathway and the common usage of ACEI (angiotensin-converting inhibitor) and ARB (angiotensin receptor blocker) drugs. ACEIs/ARBs are frequently used by patients with cardiovascular diseases and the mechanism here is linked to ACE2 downregulation by the virus, restricting its organ protective impact and resulting in RAAS (renin-angiotensin-aldosterone system) dysfunction [10]. However, this association remains inconclusive with plenty of room for further investigation. A Korean meta-analysis of 48,317 cases also suggests that pre-existing heart and vascular damage, and risk of hypercoagulability states in post-stroke patients, intensifies the cytokine-storm-induced organ damage [11]. This phenomenon of uncontrollable inflammatory reactions usually presents in advanced stages of COVID-19 infection and in the background of cardiac pathologies results in higher mortality rates in both young and elderly patients. All things considered, we suggest that pre-existing cardiovascular diseases are not only the most common comorbidities among COVID-19 patients, but also play a consequential role in unfavorable outcomes, even though the underlying mechanisms of this relationship remain a matter of discussion.

### 3.3. Metabolic Syndrome and COVID-19

Metabolic syndrome seems to be another major risk factor that influences the COVID-19 course. Components of metabolic syndrome such as hypertension, type 2 diabetes mellitus and obesity are highly predominant and crucially increase the risk of hospitalization and mortality in COVID-19 patients [12]. Many studies have shown that pre-existing type 2 diabetes mellitus (T2DM) worsens the outcomes in COVID-19 patients. In a retrospective longitudinal, multi-centered study from a cohort of 7337 confirmed COVID-19 cases in Hubei Province, China, the reported fatality rate was greater in patients with T2DM compared to non-diabetic individuals [12]. In addition, patients with T2DM were more likely to develop complications such as acute respiratory distress syndrome, acute kidney injury and septic shock. Moreover, elevated blood glucose level appears to be a key measure in modulation of the course of the disease. The risk of death (1.1% versus 11.0%) was lower in the subgroup with blood glucose <7.5 (5.2–7.5) mmol/L compared with the poorly controlled blood glucose group >7.6 (7.6–14.3) mmol/L [12].

Since diabetes has been identified as the second most common comorbidity among fatal cases of COVID-19, there have been numerous investigations to find reasons and mechanisms explaining why [12]. Enhanced ACE2 expression is believed to be related to high incidence rates in diabetic patients and recently it has been shown that ACE2 is also the cellular entry point for the virus SARS-CoV-2 [13]. Another important connection between T2DM and COVID-19 is chronic inflammation and its essential marker interleukin-6 (IL-6) as well as lymphocytes. One study from Slovenia’s University of Maribor revealed that in a group of non-survivors the serum values of IL-6 were 2–5 times higher compared with the values measured in survivors and the lymphocyte count was noticeably lower, by 2–5 times in non-survivors [13]. Obesity, the second component of metabolic syndrome, is profoundly related not only with constant chronic inflammation, but also with COVID-19 prevalence and mortality [13]. The mechanisms involving SARS-CoV-2 virus and fatty tissue interaction appear through the immune system, as the virus entry into the cell leads to over-production of inflammatory cells, specifically fat-resident regulatory T cells, and promotion of Th17-biased immunity. These processes are partly dependent on increased inflammatory cells such as IL-6, IL-23/IL-17 and TNF-a, transforming growth factor (TGF) and other macrophage inflammatory proteins. There is one interesting recent study suggesting how vaccines are less effective in obese individuals. Pellini et al. found that healthcare workers with obesity had a significantly lower antibody titer in response to vaccination than those with normal weight (*p* < 0.001) [33]. Moreover, obese patients had significantly lower serum levels of SARS-CoV-2-specific IgG antibodies. These data indicate that lifestyle changes, which can improve metabolic health and immunity, can be a strategy reducing the risk of complications and severe illness from this viral infection [32]. In addition to this, a very important field of research proposes that various types of viruses utilize fatty tissue as a reservoir, including adenovirus Ad-36, influenza A virus and HIV. Based on these facts, COVID-19 tissue and cellular localization may be interacting with fatty tissue similar to other mentioned viruses [15]. Moreover, recent studies suggest that increased body mass index (BMI) is linked with poor COVID-19 prognosis. Among diagnosed COVID-19 patients, the prevalence of individuals with obesity in hospitalized patients was much higher than that in non-hospitalized patients. For example, a study in New York City that followed 5700 hospitalized patients with COVID-19 disease revealed that 41.7% of them were individuals with a BMI > 40 kg/m^2^ [9]. Another study in France reported that obesity (BMI > 35 kg/m^2^) independently increased the risk for intensive care admission, invasive mechanical ventilation and death when compared to patients with normal BMI [12]. Physical features of individuals with obesity also likely increase COVID-19 severity and risk. Obstructive sleep apnea and other respiratory dysfunctions in patients with obesity usually increase the risk of hypoventilation-associated pneumonia, pulmonary hypertension and cardiac stress. Large waist circumference and greater body mass increase the difficulty of care in a hospital environment for supportive therapies, such as intubation, mask ventilation and prone positioning to help reduce abdominal tension and increase diaphragm capacity [14]. Metabolic syndrome is definitely a risk factor that impacts COVID-19 development and prognosis and many studies have demonstrated the importance of the care for these individuals in relation to prophylaxis, monitoring and treatment [16].

### 3.4. Hematologic Diseases and COVID-19

Despite the lack of data currently available on COVID-19 outcomes in patients with hematologic diseases, a nationwide case analysis from 575 hospitals in China suggested that, in general, immunosuppressive patients were more likely to contract COVID-19 and, most importantly, there was a higher risk of severe events among those patients, including admission to ICUs, need for assisted ventilation and number of death cases [4]. In the spring of 2020, the European Hematology Association presented several issues that challenge the managing of hematologic patients due to a changed work environment, personnel sicknesses, shortages of blood products and patients not receiving adequate therapy within the required time frame in an emergency setting [17]. The EHA experts also suggested that despite the absence of specific data, potential risk factors for severe courses of SARS-CoV-2 infection can be severe immunodeficiency, lymphopenia and long and profound neutropenia. A prospective study of a large referral hematology center in Rome, Italy, also addressed the poor representation of hematologic diseases in the comorbidity reports from patients with COVID-19. Their study, consisting of 2513 patients, both outpatients and hospitalized, found that the prevalence of COVID-19 infection in their hematologic patients (mainly affected by malignancies) was not significantly higher compared to that in the general population [18]. However, a similar case analysis study from Wuhan, China, found a 10% case rate of COVID-19 amongst 128 hospitalized patients with hematological cancer, much higher than that reported for hospitalized patients with other cancers [19]. Chinese data also show a more severe course of infection and higher case fatality rate among those patients. The increased case fatality rate for patients with hematological cancer was linked with a higher probability of decreased granulocyte concentrations because of their disease or therapy thereof [19]. Similar findings were discovered in a French study, published at about the same time. It highlighted that patients in a Hematology Department Center in Paris with a hematologic malignancy were more likely to develop a severe form of COVID-19 with acute respiratory distress syndrome, requiring mechanical ventilation, compared to those in the general French population without an underlying medical condition [20]. Higher mortality rates among those patients were also observed.

A recently published meta-analysis by Liu and co-authors suggests that a person’s blood type could also play a part in survival of COVID-19 infection and relate to the outcomes of it [31]. Five studies in their analysis provide statistically significant results, claiming that blood group A was linked to a higher COVID-19 mortality rate compared with the other blood groups. As suggested by Liu et al., why individuals with a blood group A seem to be more vulnerable to COVID-19 infection remains a matter of discussion, as only a few articles were included in the study and new evidence came into view.

Overall, patients with hematologic malignancies appear to be very vulnerable to COVID-19 infection and present a worst-case scenario with likely unfavorable outcomes. The insufficient representation of hematological diseases in the reports among common COVID-19 comorbidities remains a problem, as there are many unanswered questions left towards better disease management in the background of the pandemic and most addressed opinions are not evidence-based.

### 3.5. Kidney Diseases and COVID-19

As mentioned earlier, SARS-CoV-2 infection interacts with ACE2 receptors in renal tubules, mostly found in kidneys and lungs. After conducting postmortem biopsies there are several warnings about the impact of the virus on the cells. Macroscopically seen urinary abnormalities including proteinuria and hematuria and microscopically observed evidence for lymphocyte or macrophage infiltration next to acute tubular necrosis, together with positive viral antigen nucleoprotein, are the main findings supporting the diagnosis of acute kidney injury [22]. The damaging pathways in this case start in the lungs and following hemodynamic changes, low cardiac output and increased intrathoracic pressure can lead to acute kidney injury [21].

Since the COVID-19 outbreak, a number of studies have gathered information about novel respiratory infection consequences relating to the renal system. Several meta-analyses have revealed the AKI incidence rate in COVID-19 patients. In one of the analyses, where 24,377 patients were investigated, the results showed a 15% rate of acute kidney injury in patients with SARS-CO-V-2 infection who previously had no history of renal damage or illness. The mortality rates between patients with and without AKI were also compared. Numbers have shown tremendous differences; patients without AKI have a 12.9% mortality rate and there is a 63.1% mortality rate among patients with AKI [23].

Among the increasing numbers of people suffering from COVID-19 infection, some of those patients were had chronic renal failure requiring hemodialysis several times a week. A group of Italian doctors collected information on dialysis patients and suggested three main goals for how to reduce the SARS-CoV-2 virus in this population [24]. Those goals consist of patients, the team and dialysis ward protection. Italians advised that patients be asked to wear masks through all dialysis sessions, and patients with novel virus symptoms should be treated as COVID-positive patients until a negative PCR result; if a patient result is positive it is recommended to hospitalize them until recovery because of sudden life-threatening conditions [24]. People on dialysis have a weaker immune system response and this makes it harder to fight infections even such as a novel virus. COVID-19 is a new virus and analysis of it has only just begun. It is important to devise ways of decreasing the chance of contracting COVID-19 or making it less severe. 

### 3.6. Unmodified Factors and COVID-19

Earlier in the article we discussed the various most prevalent comorbidities having a significant impact on the outcomes of COVID-19 patients. However, despite the unquestionable influence that other conditions and pathologies owe, there are also factors that could not easily be modified but yet play a significant role in the course of COVID-19 infection. Several meta-analyses have revealed numerous significant aspects that are associated with severe COVID-19 disease progression and mortality. Rates of COVID-19 infections in women and men are similar, but the majority of COVID-19-related deaths are in men. The National Center for Health Statistics recently reported data collected from February to April 2020, which includes 37,308 deaths in the United States, revealing that 59% of patients were men. Similar statistics were published in Italy, China and South Korea, with more than 50 percent of deaths in men [26]. 

One of the articles from Beijing Tongren Hospital about sex differences in patients with COVID-19 compares the severity of COVID-19 infection in females and males, concluding that males are more susceptible and face a higher risk of worse outcomes [27]. The World Health Organization has published guidelines on how to classify confirmed cases into four categories: mild, moderate, severe and very severe. Based on the classification, men are inclined to develop more serious categories than women [28]. Furthermore, the public data from China revealed that in the first 37 cases of patients who died from COVID-19 disease, 70.3% were men and 29.7% were women. The number of men was 2.4 times higher than women in the deceased patients. An interesting fact found in a study by Jin et al. was that men and women had the same susceptibility, but men were more prone to experiencing lethal outcomes (*p* = 0.016) [34].

There are some mechanisms that are crucial to understanding how differently men and women are affected during COVID-19 infection. The major role here is played by the X-chromosome, because it is the home of many immune-associated genes [35]. For example, during bacterial or viral infections, female immune cells respond faster and more strongly than those in males, generating higher amounts of interferons, which are natural proteins that block viruses from replicating and produce antibodies to neutralize the intruders [28]. In addition to these gene-related effects, sex hormones play a crucial role in the manifestation of infectious diseases. Experiments with mice have disclosed that endogenous estrogens can ameliorate the severity of influenza infections by reducing chemokine and pro-inflammatory cytokine release, including interferon (IFN), tumor necrosis factor-α (TNFα) and C-C chemokine ligand-2 (CCL2) [27]. These findings suggest that high levels of testosterone inhibit the immune system, while estrogen facilitates immune boosting, possibly resulting in more males being affected by SARS-CoV-2 [28].

However, even though sex is an unmodifiable factor, certain sex-related behavioral and lifestyle patterns influencing susceptibility and vulnerability to COVID-19 infection could be widely recognized. Behavioral factors having an impact on the course of COVID-19 disease include lifestyle aspects such as smoking, alcohol consumption and hand-washing habits, which may contribute to a higher infection rate among males than in females [28]. Globally, men smoke nearly five times as much as women, making them even more susceptible to COVID-19. Due to reduced lung capacity or pre-existing lung disease like COPD, smokers are more susceptible and once infected, are more vulnerable to lung infections [35]. Although a systematic review in China by Guan et al. did not show any significant association between smoking and COVID-19, a multivariate analysis disclosed a 14-times increased risk of severe forms of COVID-19 disease in patients with smoking history [28]. Alcohol consumption weakens the immune system and the body’s ability to fight infection, which could enhance vulnerability to COVID-19 disease. According to the WHO, global alcohol consumption is assigned to an estimated 2.3 million deaths in males and 0.7 million in females [36]. As males consume more alcohol than females, we come to a consideration that they are also in an increased risk group for severe COVID-19 disease and death. Another important lifestyle factor that may contribute to COVID-19 infection rates is hand-hygiene habits. The WHO announced that proper hand hygiene as well as the use of personal protective equipment are the most important and powerful methods to reduce the transmission of infectious diseases, including COVID-19 [37]. One of the studies named PLACE-19, performed in Poland, analyzed the personal handwashing habits of secondary-school students aged 15–20. A total of 2323 school students were included in this study regarding their protective routine during the COVID-19 pandemic. Comparing the collected data between males and females, the study revealed that females declared a higher daily frequency of handwashing than males (*p* < 0.001). In addition, comparing respondents, females more often declared always washing their hands when necessary (68.2% vs. 54.1%; *p* < 0.001), while males indicated various reasons for not handwashing [29]. Another study conducted in highway service station restrooms in England reported that only 31% of men washed hands and/or used soap compared to 65% of women. This difference between males’ and females’ personal hand washing hygiene could be related to more COVID-19 cases in males [28]. Many studies have demonstrated that age is a risk factor for poor COVID-19 outcomes. Analyses of COVID-19 cases in China, the United Kingdom and Italy consistently demonstrate that there is a correlation between age and hospitalization, ICU admission, recovery time and fatality [29]. Another study from the Shanghai Public Health Clinical Center found that COVID-19 patients aged ≥60 years had a higher rate of respiratory failure and needed more prolonged treatment than those aged <60 years. Moreover, the cure rate for patients aged ≥60 years (89.4%) was relatively lower than for patients aged <60 years (95.6%), especially in males. These findings reveal that elderly COVID-19 patients had much more severe disease and showed poorer response to treatments than the younger age group [30].

## 4. Limitations

Findings in this literature review are affected by several limitations. First of all, although we have systematically searched the literature to discover suitable articles, it is possible that some studies might have been missed. Regardless of the wide-ranging search approach, non-English-language studies may not have been listed in the databases we searched. Secondly, having in mind that the majority of data are from China, the results should be compared to other countries with caution as the prevalence of comorbidities varies remarkably among different populations and geographic regions. We are also aware that there are many more conditions and comorbidities possibly having a significant impact on the course of infection. However, the aim of this review was to discuss issues related to the most prevalent and highly impactful factors. Another element that may affect the analysis of the data is varying sample size in the different studies, which can be an important limitation for reaching statistical competence. Lastly, due to the rapidly evolving situation of COVID-19, publications with new data will appear, and future studies with larger sample sizes will likely give us more information about additional comorbidities.

## 5. Conclusions

Throughout the two years of the pandemic, COVID-19 has taught us important lessons about its features, clinical signs and symptoms, treatment, outcomes and, as we have discussed in our article, the interplay with comorbidities. Based on the recent studies, patients with comorbidities face a higher risk for worse outcomes and increased mortality. Critical situations develop in individuals with heart diseases, immunosuppression, kidney diseases, metabolic syndrome and hematologic diseases. Unmodified factors like older age and male sex also present increased risk for severe events. Hospital workers must take these findings into consideration and manage patients with comorbidities accordingly. Although the world after the pandemic will never be the same, the knowledge that was gained during it will change tremendously the perception of this disease in relation to patients with underlying conditions. 

## Figures and Tables

**Figure 1 ijerph-19-04738-f001:**
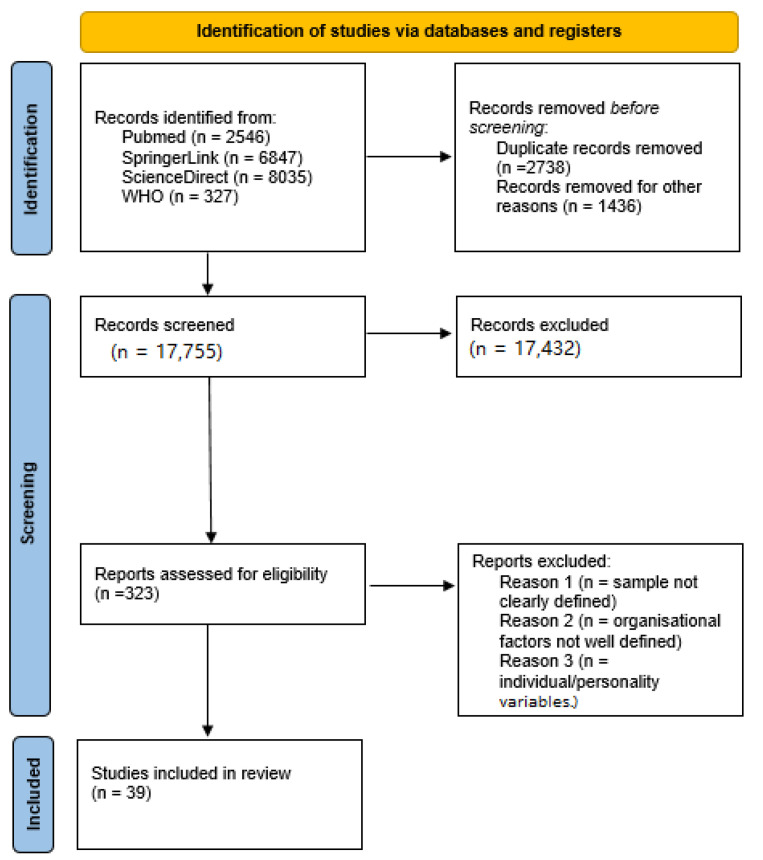
Flow chart.

**Table 1 ijerph-19-04738-t001:** Summary of the selected studies for the mini review.

Investigator (Year)	Location	Study Design	Number of Patients	Study Population	Findings
Liu et al., 2020 [3]	China	Review	2007	COVID-19 and cancer	Might impact cancer diagnosis
Liang et al., 2020 [4]	China	Cohort	18	COVID-19 and cancer	Patients with cancer—severe consequences
Gosain et al., 2020 [5]	USA	Review	53	COVID-19 and immunosuppression	Remdesivir may have clinical benefits
Ruiz et al., 2020 [6]	Spain	Observational study	18	Coronavirus after solid organ transplant	SARS-CoV-2 infection has a severe course in SOT patients
Wu et al., 2020 [7]	China	Viewpoint	72,314	Key findings from diagnosed cases	Characteristics of COVID-19 outbreak
Grasselli et al., 2020 [8]	Italy	Investigation study	1591	Characteristics of COVID-19 in ICU	Mechanical ventilation and mortality
Huang et al., 2020 [9]	China	Observational study	41	COVID-19 clinical features	COVID-19 was associated with high mortality
Pranata et al., 2020 [10]	Indonesia	Meta-analysis	4448	Cardiovascular diseases and COVID-19	Increased risk for poor outcomes
Bae et al., 2021 [11]	South Korea	Meta-analysis	48,317	COVID-19 and hypertension + diabetes	Increased risk of fatal outcomes
Bansal et al., 2020 [12]	USA	Review	63,636	Metabolic syndrome and COVID-19	Poor outcomes and increased mortality
Marhl et al., 2020 [13]	Slovenia	Observational study	82,511	COVID-19 and diabetes	Higher risk for COVID-19
Popkin et al., 2020 [14]	USA	Review	399,461	Obesity and COVID-19	More at risk for COVID-positive
Petrakis et al., 2020 [15]	Romania	Review	-	Obesity and COVID-19	Negative prognosis for obese people
Costa et al., 2020 [16]	Brazil	Review	108,287	Metabolic syndrome and COVID-19	MS is a risk factor
Lilienfield-Toal et al., 2020 [17]	Germany	Review	-	SARS-CoV-2 in cancer patients	Increased risk for worse outcome
Girmenia et al., 2020 [18]	Italy	Observational study	2513	COVID-19 and hematologic disorders	No significant correlations
He et al., 2020 [19]	China	Cohort	354	COVID-19 and hematological cancers	Severe COVID-19 and more deaths
Malard et al., 2020 [20]	France	Review	25	COVID-19 and hematologic disorders	Higher incidence of severe events
Hassanein et al., 2020 [21]	USA	Review	2072	COVID-19 and kidneys	Acute kidney injury (AKI) is common in COVID-19
Farouk et al., 2020 [22]	USA	Review	8776	COVID-19 and kidneys	AKI increased mortality
Shao et al., 2020 [23]	China	Meta-analysis	24,527	AKI and COVID-19	AKI is associated with higher mortality rates
Rombola et al., 2020 [24]	Italy	Review	60	COVID-19 and dialysis patients	Accelerated healthcare protocols
Jutzeler et al., 2020 [25]	Switzerland	Meta-analysis	12,149	Comorbidities and COVID-19	Comorbidities are associated with severity and mortality
Bienvenu et al., 2020 [26]	Australia	Review	17,000,000	COVID-19 and males	Male sex is a risk factor
Jin et al., 2020 [27]	China	Review	1623	COVID-19 and sex differences	Males with COVID-19 are at risk for worse outcomes
Acharya et al., 2020 [28]	Ireland	Review	44,672	Gender disaggregation in COVID-19	Higher COVID-19 susceptibility in males
Guzek et al., 2020 [29]	Poland	Observational study	2,170,464	COVID-19 and personal behaviors	Differences between gender
O’Brien et al., 2020 [30]	Canada	Review	101,121	Impact of age and sex on COVID-19	Male sex and older age—worse outcomes
Liu et al., 2020 [31]	China	Comment	221	Age and COVID-19	Prognosis was worse in patients older than 60 years
Pellini et al., 2021 [32]	Italy	Observational study	248	COVID-19 and vaccination	Correlation of BMI classes with antibody titres
Ealey et al., 2021 [33]	Canada	Review	900,000	COVID-19 and obesity	Vaccines are less effective in obese individuals

**Table 2 ijerph-19-04738-t002:** Summary of the main results related to comorbidities.

Comorbidities	Results	References
**Immunosuppression**	Cancer patients: higher risk for unfavorable and lethal outcomes, more likely to be admitted to ICU and receive invasive ventilation; Organ recipients: high percentage develop respiratory deficiency; hard to manage with additional comorbidities.	Liu et al., 2020 [3] Liang et al., 2020 [4] Gosain et al., 2020 [5] Fernandez-Ruiz et al., 2020 [6] Wu et al., 2020 [7]
**Cardiovascular**	Suggestion that hypertension is the most common comorbidity among COVID-19 patients; severe forms of COVID-19; high fatality rate; requiring aggressive ventilation at the ICU.	Wu et al., 2020 [7] Grasselli et al., 2020 [8] Huang et al., 2020 [9]
**Metabolic syndrome**	Increased hospitalization rates among diabetic and obese patients; acquire complications such as respiratory distress, AKI, septic shock; poorly controlled diabetes worsens course of the disease, higher fatality rate.	Bansal et al., 2020 [12] Marhl et al., 2020 [13] Costa et al., 2021 [16]
**Hematologic**	Higher risk of severe events, need of ventilation, higher fatality rate; severe COVID-19 development.	Liang et al., 20202 [4] von Lilienfeld-Toal et al., 2020 [17] Girmenia et al., 2020 [18]
**Renal**	High AKI incident rate; dialysis patients hard to manage and protect from COVID-19.	Shao et al., 2020 [23] Rombola et al., 2020 [24]
**Unmodified factors**	Higher percentage of COVID-19 related to men; men develop severe forms of disease; no correlation between smoking and COVID-19, patients >60 years need prolonged treatment, higher rate of respiratory failure.	Bienvenu et al., 2020 [26] Jin et al., 2020 [27] O’Brien et al., 2020 [30]

## Data Availability

Not applicable.

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
