# Peer review of "Mini Review: Co-Existing Diseases and COVID-19—A One Way Ticket?"

_ijerph, 2022, doi:10.3390/ijerph19084738_

Round 1

Reviewer 1 Report

The article deals with an important topic, even though the research is gathering data about previous studies related to Covid and comorbidities. The authors mentioned the limitations of this study, which is a healthy thing to do nowadays. The results as a summary are valuable. Some aspects need to be improved before publication.

As the authors mentioned co-morbidities affect the covid disease outcome.  Please explain in the manuscript, how the data is gathered when for instance an individual has obesity and hypertension and dies. How this data is registered? Are both co-morbidities considered? Every reported article use the same methodology for registering these cases? For instance the reason for death could be hypertension or obesity or both?

There are several typos and grammar errors. Please check the whole manuscript.

Line 227. The authors claim the pre-existing cardiovascular problems are the most common co-morbidities. This needs statistic support or change the claim to suggest.

Line 144. Please provide reference about the claim that diabetes is the second most common co-morbidity among death cases.

Line 154. Provide support to the claim.

Line 220. There are some studies that relate blood type to covid outcomes. Please add some references for that.

Line 283. Provide references to support this claim.

In summary, the article deals with an important topic. I think a summary Table would help to highlight the main results of this article related to comorbidities.

Author Response

Thank you for your review.
1)As the authors mentioned co-morbidities affect the covid disease outcome.  Please explain in the manuscript, how the data is gathered when for instance an individual has obesity and hypertension and dies. How this data is registered? Are both co-morbidities considered? Every reported article use the same methodology for registering these cases? For instance the reason for death could be hypertension or obesity or both? We tried to explain everything in the results.

2)There are several typos and grammar errors. Please check the whole manuscript. Manuscript reviewed for grammar errors

3)Line 227. The authors claim the pre-existing cardiovascular problems are the most common co-morbidities. This needs statistic support or change the claim to suggest. Changed to suggest, discussing this topic based on the literature data above.

4)Line 144. Please provide reference about the claim that diabetes is the second most common co-morbidity among death cases. Reference provided

5)Line 154. Provide support to the claim. Reference provided

6) Line 220. There are some studies that relate blood type to covid outcomes. Please add some references for that.  Discussion updated with results of meta-analysis on this topic.

7)Line 283. Provide references to support this claim. Reference provided.

8) In summary, the article deals with an important topic. I think a summary Table would help to highlight the main results of this article related to comorbidities. Summary of the main results related to comorbidities presented in FIgure 3.

Reviewer 2 Report

This paper focuses on the analysis of SARS-CoV2 organic involvement and preexisting comorbidities. The aim was to evaluate the impact of previous disease on COVID prognosis. 

[Line 11]: Please, correct the following sentence: " Their presence in the background of corona virus [coronavirus] show a tendency to worsen the course of 11 disease and increase the risk of unfavorable outcomes."

[Lines 19-20]: Please, correct the following sentence: "Metabolic syndrome and its’ [its] components has been identified as the second 19 most common comorbidity among fatal cases of COVID-19." Otherwise, please change the last sentence, because it sounds too similar to "patients [3] and has been identified as the second most common comorbidity among cases of COVID-19 [26]." [Rico-Martín S, Calderón-García JF, Basilio-Fernández B, Clavijo-Chamorro MZ, Sánchez Muñoz-Torrero JF. Metabolic Syndrome and Its Components in Patients with COVID-19: Severe Acute Respiratory Syndrome (SARS) and Mortality. A Systematic Review and Meta-Analysis. J Cardiovasc Dev Dis. 2021;8(12):162. Published 2021 Nov 25. doi:10.3390/jcdd8120162]

[lines 40-42]: "Our university hospital physicians who worked directly in the COVID-19 dedicated wards and Intensive care units. It was observed, that in the most cases, pre-existing comorbidities played a huge role in patient’s course of disease and unfavorable outcomes. ". Ok, I'm sorry, I don't understand the meaning. I think there are errors in the grammatical construction of the sentence. [maybe, you can put a dot.]

[Line 67] please, provide a specific "result" section in which you can resume your results and a "discussion" section where do you criticized the results. 

In your discussion, consider to distinguish the COVID cardiovascular clinical manifestations and the impact of the preexisting comorbidities. 

For example, for SARS-CoV2 cardiac involvement consider to discuss the clinical and pathological manifestations of COVID in contrast with previous diseases, for example citing: 

Maiese A, Frati P, Santoro P, Manetti AC, La Russa R et al Myocardial Pathology in COVID-19-Associated Cardiac Injury: A Systematic Review. Diagnostics (Basel). 2021 Sep 8;11(9):1647. doi: 10.3390/diagnostics11091647. PMID: 34573988; PMCID: PMC8472043.

At same times, for the "Hematologic diseases and COVID-19" there is the great problem of the hypercoagulability. In this case you can expand your discussion adding this kind of studies:

Zanza C, Racca F, Longhitano Y, et al. Risk Management and Treatment of Coagulation Disorders Related to COVID-19 Infection. Int J Environ Res Public Health. 2021;18(3):1268. Published 2021 Jan 31. doi:10.3390/ijerph18031268

If you do a Systematic Review, please, follow the PRISMA guidelines.

Please, provide many tables in a supplementary file. In those tables you can summarize your results in a more comprehensible way. 

Moreover, please, read the latex-form (Word) provided by MDPI: on the references section is demanded to insert DOI.

Hope it was helpful for your good work, 

Sincerely

Author Response

1) [Line 11]: Please, correct the following sentence: " Their presence in the background of corona virus [coronavirus] show a tendency to worsen the course of 11 disease and increase the risk of unfavorable outcomes." Corrected

2)[Lines 19-20]: Please, correct the following sentence: "Metabolic syndrome and its’ [its] components has been identified as the second 19 most common comorbidity among fatal cases of COVID-19."  Corrected

3) [lines 40-42]: "Our university hospital physicians who worked directly in the COVID-19 dedicated wards and Intensive care units. It was observed, that in the most cases, pre-existing comorbidities played a huge role in patient’s course of disease and unfavorable outcomes. ". Ok, I'm sorry, I don't understand the meaning. I think there are errors in the grammatical construction of the sentence. [maybe, you can put a dot.] - Grammar error corrected

4) [Line 67] please, provide a specific "result" section in which you can resume your results and a "discussion" section where do you criticized the results. Results and discussion section added.

5) For example, for SARS-CoV2 cardiac involvement consider to discuss the clinical and pathological manifestations of COVID in contrast with previous diseases, for example citing. This study focused on states caused by COVID-19, we selected studies about pre-existing comorbidities in COVID-19 patients.

6)At same times, for the "Hematologic diseases and COVID-19" there is the great problem of the hypercoagulability. In this case you can expand your discussion adding this kind of studies:  we tried to select studies about the patients with comorbidities facing COVID-19, this article seems to focus about hypercoagulability problems caused by coronavirus, not on pre-existing hypercoagulability states.

7) Please, provide many tables in a supplementary file. In those tables you can summarize your results in a more comprehensible way. Summary table added

8) Moreover, please, read the latex-form (Word) provided by MDPI: on the references section is demanded to insert DOI. Reference section corrected, DOI inserted.

Reviewer 3 Report

Authors aims to analyze the role of a certain underlying comorbidities in COVID-19 patients regarding their influence to unfavorable and fatal outcomes.

The aim of this review should not be to analyze, given that there is no meta-analysis tools involved, authors are reviewing results under comorbidities conditions.

The extension of the manuscript and therefore the content, in terms of used references would make it more like a mini-review than a full review.

Specific comments/suggestions are highlighted in the attached .pdf file.

The document need correction on the specific suggestion, but also it could help to use conventional section names (Results, Discussion or Results and Discussion) in other to make easier to follow-up for reader. The topic is of high relevance, but when authors talks about the relative risk measures, this should be declared (e.g. OR = 12) and include 95% CI reported in the original studies plus the p value.

Author Response

The aim of this review should not be to analyze, given that there is no meta-analysis tools involved, authors are reviewing results under comorbidities conditions.- correction of the aim, which would represent the manuscript correctly, was made

The extension of the manuscript and therefore the content, in terms of used references would make it more like a mini-review than a full review.

Specific comments/suggestions are highlighted in the attached .pdf file. – all specific comments corrected in the original manuscript

The document need correction on the specific suggestion, but also it could help to use conventional section names (Results, Discussion or Results and Discussion) in other to make easier to follow-up for reader. The topic is of high relevance, but when authors talks about the relative risk measures, this should be declared (e.g. OR = 12) and include 95% CI reported in the original studies plus the p value. – results and discussion section was created so the manuscript would be understood in more clear way, how the reviewed work impacts on the aim of the study and how conclusions are related to the results.

Round 2

Reviewer 1 Report

The authors improved the paper .

Reviewer 3 Report

Authors aims to review the role of a certain underlying comorbidities in COVID-19 patients regarding their influence to unfavorable and fatal outcomes.

The authors incorporated the suggestions and comments made by this reviewer in a satisfactory manner. Consideration is appreciated and significant improvements are noted in the development of the manuscript, making its central idea clearer and easier to follow.